# Intrauterine Transmission of Zika and Vertical Transfer of Neutralizing Antibodies Detected Immediately at Birth in Oaxaca, Mexico: An Analysis in the Context of Microcephaly

**DOI:** 10.3390/microorganisms12030423

**Published:** 2024-02-20

**Authors:** Alfredo Porras-García, Dina Villanueva-García, Rafael Arnaud-Rios, Nadia García-Lemus, Angélica Castillo-Romero, Mariana Mejía-Flores, Luis Erik Contreras, Liliana Hernández-Castillo, Elva Jiménez-Hernández, Juan Manuel Mejía-Aranguré, Sara A. Ochoa, Juan Xicothencatl-Cortes, Ariadnna Cruz-Córdova, Rosalia Lira-Carmona, José Arellano-Galindo

**Affiliations:** 1Unidad de Cuidados Intensivos Neonatales, Hospital Infantil de México Federico Gómez, Mexico City 06720, Mexico; alfredoporras21@gmail.com (A.P.-G.); dinavg21@yahoo.com (D.V.-G.); nadia_ame44@hotmail.com (N.G.-L.); angie_carom@hotmail.com (A.C.-R.); 2Hospital General de Pochutla Oaxaca, San Pedro Pochutla 70900, Mexico; rafael.ar1980@gmail.com; 3Unidad de Investigación en Enfermedades Infecciosas, Laboratorio de Virología Clínica y Experimental, Hospital Infantil de México Federico Gómez, Mexico City 06720, Mexico; mariana_panic81@hotmail.com (M.M.-F.); jose.arellano@gmail.com (L.E.C.); hdzcastle@gmail.com (L.H.-C.); 4Hospital Pediátrico Moctezuma SEDESA, Universidad Autónoma Metropolitana, Mexico City 15530, Mexico; elvajimenez@yahoo.com; 5Instituto Nacional de Medicina Genómica, Mexico City 14610, Mexico; arangure.jm@gmail.com; 6Laboratorio de Bacteriología Intestinal, Hospital Infantil de México Federico Gómez, Mexico City 06720, Mexico; saraaridnah@hotmail.com (S.A.O.); juanxico@yahoo.com (J.X.-C.); ariadnnacruz@yahoo.com.mx (A.C.-C.); 7Unidad de Instigación Médica en Enfermedades Infecciosas y Parasitarias, Hospital de pediatría Centro Médico Nacional, Siglo XXI, Instituto Mexicano del Seguro Social, Mexico City 06720, Mexico; 8Centro Interdisciplinario de Ciencias de la Salud, Departamento de Medicina, Unidad Milpa Alta, Instituto Politécnico Nacional, Mexico City 12000, Mexico

**Keywords:** vertical transference, ZIKV, viremia, virolactia, DBS, pair infant–mother

## Abstract

Zika virus (ZIKV) can cause neurological issues in infants. To provide protection, neutralizing antibodies should be transferred from the mother to the infant. We conducted a study at the Hospital General de Pochutla, Oaxaca, Mexico. Samples were collected from mothers (blood and breast milk) and infants (saliva and dried blood spots) within the first 12 postnatal hours (December 2017 to February 2018) and tested for ZIKV total and neutralizing antibodies as well as ZIKV-PCR. Microcephaly was evaluated according to INTERGROWTH-21st standards. Maternal IgG seroprevalence was 28.4% with 10.4% active infection, while infant IgG seroprevalence was 5.5% with 2.4% active infection. There were two cases of virolactia, and 6.3% of the infant saliva samples tested positive for ZIKV. Additionally, 18.3% of the infants were in a cephalic perimeter percentile lower than 10 and had an association between microcephaly and serology or a PCR between 8.6 and 60.9%. The infant blood samples had neutralizing antibodies, indicating intrauterine protection. Microcephaly was correlated with serology or PCR, but in our study population, non-ZIKV factors may be involved as well. Low ZIKV infection values in breast milk mean that breastfeeding is safe in most of the mothers and infants of the endemic area studied.

## 1. Introduction

Zika virus (ZIKV) is an arbovirus member of the *Flaviviridae* family (i.e., an enveloped, icosahedral virus with positive-sense single-stranded RNA) [1]. It was first identified in Uganda in 1947 and is recognized to have caused a human outbreak in 2007 in Micronesia [2,3]. Before that event, only 14 cases had been reported [4]. Human ZIKV infection is transmitted by bites from the female *Aedes* mosquito [2]. About 80% of infections are asymptomatic, but it can present as a maculopapular rash, fever, arthralgia, and conjunctivitis. Flaccid paralysis, resembling Guillain–Barre, also occurs in 0.24 cases per 1000 [5,6,7,8]. Other methods of ZIKV transmission are also known, such as congenital vertical transmission. The most significant risk for the development of congenital Zika syndrome at birth is the fetus acquiring the infection during the first or at the beginning of the second pregnancy trimester, concurrent with major ongoing neurodevelopment [5,6]. Maternal ZIKV infection can lead to neonatal congenital Zika syndrome (CZS). While microcephaly is the most common manifestation, other neurological malformations like intracranial calcification, brain atrophy, cortical and corpus callosum abnormalities, ventriculomegaly, hydranencephaly, and ocular defects also occur [7,8,9]. The presence of ZIKV antibodies at birth is a form of passive immunization that the mother transfers to the infant, which represents protection against the infection acquired by the bite of a mosquito once the extrauterine life has begun [10,11]. In this context, it is important to know the neutralizing ability of the IgG ZIKV antibodies transferred from mother to newborn, mainly in endemic zones of flavivirus. Herein, we propose to determine the neutralizing activity of the anti-Zika antibodies transferred vertically from mother to infant and correlate the presence of such antibodies with microcephaly at birth, transferred to infants from mothers with positive serology, in an endemic zone of flavivirus in Mexico.

## 2. Materials and Methods

### 2.1. Study Design

This was an observational, cross-sectional study of all mother–infant dyads attended by the obstetric department of the Hospital General de Pochutla, Oaxaca, within the first 12 postnatal hours during the study period (December 2017 to February 2018). This study period was considered optimal because it included dyads who had been in their first or second pregnancy trimester during the rainy season, when the risk of exposure to this vector-borne disease is most common [12,13]. Clinical data were collected and infant head circumference was assessed based on INTERGROWTH-21st values [14]. The experimental methodology is summarized in Figure 1.

### 2.2. Study Protocol and Samples

The study was approved by the Research, Biosafety and Ethics Committee at the Hospital Infantil de México Federico Gómez (HIM/2017/015.1337 and HIM/2018/037.1512). The Hospital General de Pochutla, Oaxaca, was selected as the study site because it serves the population of the coastal zone and low mountain range in Oaxaca, which is considered endemic for vector-borne diseases [15]. Maternal informed consent was obtained prior to study participation. Samples of 3 mL of maternal EDTA–blood and 2 mL of breast milk (BM) were obtained, as was a mouth swab in 2 mL of transport medium and blood in spots (DBS) samples from the infant within the first 12 postnatal hours. All samples were transported immediately in less than 48 h from Pochutla, Oaxaca, to Mexico City for processing; during the transportation, the samples were stored at 4 degrees, and DBS was packed in a sterile plastic pack at room temperature until the transportation.

### 2.3. Serology

Maternal and neonatal ZIKV IgG and IgM were analyzed with ELISA according to the manufacturer’s instructions. IgG was prepared with the MyBioSource Zika IgG kit (cat. MBS 109002, San Diego, CA, USA). In brief, serum samples from the mother were diluted 1:10 in a dilution buffer, and sample and controls were added to each well followed by HRP-conjugated. The wells were incubated at 37 °C and then washed 4 times. Afterward, a chromogen solution was added and the samples were incubated for 15 min at 37 °C. Finally, stop solution was added and read at 450 nm in an ELISA reader. The maternal IgM was prepared as above but with the MyBioSource Zika IgM kit (cat. MBS109003, San Diego, CA, USA). The ELISA IgG analysis was performed on the DBS of the newborns with the EUROimmune kit (cat EI2668-9601G, Lübeck, Germany) according to the previously published procedure [16].

### 2.4. RT-PCR

We used 500 µL of whole blood and 500 µL fresh BM from the mother, the latter centrifuged to 9000× *g* for 10 min. Neonatal blood was recovered from DBS as described [16,17,18], and 100 µL of saliva was recovered in transport medium. RNA was obtained from each sample using TRIzol^®^ (cat. 15596026, Invitrogen Thermo Fisher Scientific, Carlsbad, CA, USA). In brief, 500 µL from each sample was placed in microtubes with 500 µL of TRIzol, following the manufacturer’s recommendation. Amplification was performed with previously described primers and probes from the TaqManTMRNA-to-CtTM 1-Step Kit (cat. 4392653, Thermo Fisher Scientific) [19].

### 2.5. TORCH

Panel: Anti-TORCH IgG and IgM antibodies were analyzed in maternal serum and neonatal blood for all RT-PCR ZIKV-positive patients. For the latter, two 0.5 cm semicircles of DBS were recovered using three punches; then, they were placed in 200 µL of PBS 1X buffer (pH 7.4) with shaking overnight at 4 °C. IgG was detected using the EUROImmune immunoblot system (DN 2410-1601-11, EUROLine, Lübeck, Germany), which detects IgG antibodies against *Toxoplasma gondii*, rubella virus, CMV, HSV-1 and HSV-2, *Bordetella pertussis*, *Chlamydia trachomatis*, parvovirus B19, *Treponema pallidum*, and VZV. IgM was detected with the EUROImmune anti-TORCH IgM kit (DN 2410-1601-4 M, EUROLine). Both kits were used according to the manufacturer’s instructions. In brief, strips were blocked with buffer, after which maternal serum (1:51)/infant DBS (1:10) was added, and then the strips were incubated for 30 min and washed. Afterward, conjugate was added, and the strips were again incubated for 30 min and washed; then, the antibodies were detected using the kit substrate. Test evaluation was performed via EUROLineScan using the manufacturer’s recommended cutoff.

### 2.6. Titration of the ZIKV Strain

The viral strain (ATCC CCL-8, Manassas, VA, USA) was propagated in Vero cells and harvested 7 days after the infection. The suspension was centrifugated to 4000× *g* for 10 min, and afterward, the supernatant was filtered through a membrane filter of 0.45 µm (BRAND^®^ accu-jet^®^ cat. Z333913, Burlington, MA, USA), aliquoted, and stored at −80 °C. Then, a 50% tissue culture infectious dose (TCID50) was deposited in a 96-flat-well plate with 2 × 104 Vero cells. The viral strain was diluted between 10-1 and 10-8 in DMEM 2% fetal bovine serum in a quadruplicated assay. Each dilution was added on the microplate; then, this was incubated for 7 days at 37 °C/5% CO_2_ and the appearance of the cytopathic effect (CPE) was analyzed. The TCID50 was calculated through the Spearman–Kärber method, using the following equation: Log DICT50 = log(highest dilution giving 100%CPE) + 0.5 − (total number of test units showing CPE/number of test units per dilution) [20].

### 2.7. Neutralizing Assay

A microneutralizing test was standardized with the antibody Magic™ Rabbit Anti-ZIKV E Monoclonal antibody, clone ZV67 (CABT-ZS1015, CD creative Diagnostics, Shirley, NY, USA), and strain MR766 (ATCC VR-84, Manassas, VA, USA). A viral dosage of 400 DICT50 was mixed with dilutions of 1:20, 1:40, 1:80, 1:160, 1:320, 1:640, 1:1280, and 1:2560. These were incubated for 1 h/37 °C and then each dilution was added in a microplate of 96 wells with 2 × 104 Vero cells in each well. The plate was incubated for 5 days. Next, the plate was washed and fixed with cold acetone. Afterward, the plate was washed with PBS 1X, fixed with cold acetone and then permeabilized with triton 0.1% in PBS/30 min, followed by a step where the endogenous peroxidase was blocked with a solution of 0.03% hydrogen peroxide in methanol for 30 min. Afterward, the plate was washed. Then, PBS 1X with 10% fetal bovine serum was added, and the samples were incubated at room temperature for 30 min. Afterwards, the monoclonal antibody anti-ZIKV envelope monoclonal protein of mouse antibody GT363 diluted 1:50 (MA5-42346, Thermo Fisher Scientific, Waltham, MA, USA) was added and incubated for 30 min, and the plate was washed afterward. Then, a second antibody anti-mouse Goat IgG (H+L) secondary antibody 1:200, HRP (Thermo Fisher Scientific, Cat. 31430), was added and incubated for 30 min as a revealer. A solution of AEC 0.4% in dimethylformamide was added, the plates were counted, and the effective dosage of 50 (ED50) was calculated with an ED50 Calculator (AAT Bioquest, Inc. Pleasanton, CA, USA) [21,22]. DBS from the infant was hydrated as previously written [23] and treated as described above.

### 2.8. Variable Definitions

Active maternal and active infant ZIKV infections were defined as the detection of viral RNA by real-time polymerase chain reaction (PCR) in blood, regardless of the associated clinical symptoms. Congenital CZS was defined as the detection of viral RNA by PCR in dried blood spots (DBS), in the presence of birth defects associated with ZIKV infection during gestation [24]. Microcephaly was defined as an occipitofrontal circumference below the 10th percentile [25]. ED50 was defined as the concentration of DBS blood from the infant which neutralizes 50% of the viral infectivity.

## 3. Results

During the three-month study period (December 2017 to February 2018), we enrolled 126 mother–infant dyads, whose clinical and demographic characteristics are shown in Table 1.

Most mothers were over the age of 19 years (105/126, 83%); the others were adolescents (21/126, 17%). Nearly all infants were born full-term (120, 95%) (Table 1). Of the 126 individuals, 36/126 had maternal IgG+ seroprevalence (28.6%) and 39/126 had IgM+ seroprevalence (31%). Among those with IgG seroprevalence, most (28/126) had both IgG+ and IgM+ (22.2%), followed by IgG+ and IgM−, 8/126 (6.3%), followed by IgG− and IgM+, 11/126 (8.7%). Additionally, 84 of the 126 individuals (66.6%) had neither IgG nor IgM. Lastly, the infants had an IgG ZIKV seroprevalence of 5.5%. Maternal RT-PCR-based prevalence in blood was 10.3% (13/126). Regarding the maternal TORCH panel in the positive PCR samples, no participant was IgM+. However, IgG+ results were found for HSV-1 (8/13 (61.5%)), CMV (7/13 (53.8%)), rubella (3/13 (23.1%)), VZV (3/13 (23.1%)), parvovirus B19 (3/13 (23.1%)), and Toxoplasma gondii (1/13 (7.6%)). Of these, 7/25 (28%) were IgG+/IgM+, 1/6 (0.8%) were IgG+/IgG−, and 5/11 were IgG−/IgM+ (Table 2). In contrast, when ZIKV was searched for in milk, it was detected in 2/126 individuals (0.015%). In 6.3% (8/126) of the infants, ZIKV was detected in saliva, and in 1.6% (2/126), it was detected in DBS. Among those, when analyzed by PCR, 7/126 (7.9%) were IgG+/IgM+, 1/126 (3.2%) were IgG+/IgM−, and 5/126 (11.1%) were IgG−/IgM+ (Table 2).

Neutralizing activity was tested in 13 infant DBS samples. Of the samples which were PCR-positive, seven were IgG-positive in the blood from the infants. The results were as follows: In total, 9/13 (samples 61, 62, 65, 73, 76, 82, 93, 96, and 130) did show a considerable neutralizing activity. Out of them, 6/9 (samples 61, 62, 65, 73, 76, and 130) were IgG+ in the mother, and out of those, 7/9 were IgM+ as well. Additionally, one of them was virolactia-positive (sample 76). On the other hand, 4/13 did show a very low level of ED50; out of these, 2/4 were IgG−/IgM+, and 2/4 were IgG+/IgM+. ED50 in the blood of the infants was not dependent on the maternal serology. Concerning ED50 and serology in the newborn, the level of ED50 was variable; 5/7 samples were IgG-positive with high levels (samples 62, 65, 82, 93, and 96), whereas for 2/7, the ED50 level was low. Additionally, of the 6/13 IgG negative DBS samples, 4/6 did have a high DE50 level (samples 61, 65, 73, and 76), whereas 2/6 had low levels (samples 20 and 50). When the ED50 level was compared with those ZIKV-PCR saliva samples from the infant using the neutralization assay, 8/13 were positive. Of these, 6/8 did show a considerable neutralizing activity, whereas 2/8 showed a very low level of neutralizing activity. Additionally, 3/13 were DBS PCR-positive. Of these, one (sample 96) had a considerable DE50 level, whereas the others two had low neutralizing activity (samples 82 and 120) (Figure 2).

In the studied population, 23 out of the 126 newborns (17.46%) were in a percentile ≤10. Out of these 23 newborns, 9 had IgG-positive mothers, 14 had IgM−positive mothers, and 7 were blood PCR-positive. The odds ratio (OR) for IgG positivity in mothers was 1.8, with a *p*-value of 0.21. The odds ratio for IgM positivity in mothers was 4.8, with a *p*-value of 0.0011. The OR for blood PCR positivity in newborns was 1.16, with a *p*-value of 0.81. When analyzing samples from newborns in a percentile ≤10, we found that 5 out of 23 were positive for PCR in saliva, with an OR of 9.2 and a *p*-value of 0.0040. Additionally, 2 out of 23 were IgG-positive in DBS, with an OR of 2.4 and a *p*-value of 0.34 (Table 3).

Out of a total of six cases, IgG DBS ZIKV was detected by PCR in the blood of five out of six mothers and in the BM of one out of six mothers. Two out of six infants had a head circumference in a percentile of 10 or less (samples 93 and 124). In one of these cases, the maternal serology was IgG+/IgM−, and the PCR test was positive for blood but negative for BM. On the other hand, the infant was IgG-positive with a positive PCR test for saliva but negative for PCR DBS in the blood. In the second case, the maternal serology was IgG+/IgM+, and the mother tested positive for ZIKV by PCR for blood but negative by PCR for BM. Meanwhile, the infant was IgG-positive with viral detection in saliva as well as in DBS. Out of the remaining four cases (samples 50, 82, 96, and 128), the first sample (sample 93) of the two samples confirmed to have microcephaly showed an ED50 value interpreted to have good neutralization activity, whereas the second sample (sample 124) had an ED50 of 0.006 × 10^3^, interpreted as null neutralization (Table 4).

## 4. Discussion

Viruses of the *Flaviviridae* family, such as dengue, Zika virus (ZIKV), and chikungunya, can be transmitted vertically from mother to newborn. Dengue virus (DENV) can be transmitted from a mother to her infant with a transmission rate of 18.5% [26,27]. Zika virus (ZIKV) is a major concern because of its potential to cause harmful outcomes in infants, including microcephaly, intracranial calcification, and fetal death [28]. Infants living in areas where the Zika virus (ZIKV) is prevalent are at risk of contracting the infection either through mosquito bites or via BM. This is concerning because some studies suggest that ZIKV infection in newborns can affect their postnatal neurodevelopment during the first few years of their life. However, the presence of anti-ZIKV antibodies, especially those with neutralizing activity that can be transferred from mother to infant through lactation, can play an important role in reducing these risks. This study examines the link between ZIKV transmission from mother to child and microcephaly, as well as the acquisition of neutralizing antibodies by the infant from the mother.

Pochutla, Oaxaca, is one of 30 districts in Mexico’s Oaxaca state. This coastal zone has a hot, humid climate and a long rainy season, typically from May to October. This Mexican region also has a high poverty rate [29,30]. These environmental and social conditions may contribute to increased vector-borne ZIKV transmission [31,32], increasing the importance of assessing the seroprevalence of ZIKV in mother–neonate dyads in this region.

The prevalence of IgG-based ZIKV varies across different regions in the world. In French Polynesia it is recorded at 49%, while in El Salvador, Brazil, and Nicaragua it is 63% and 56%, respectively [32,33,34,35]. In Mexico, the first case of ZIKV was reported in June 2015, and the prevalence rate reached 30.8%, whereas in a pregnant sample, it was estimated to be 36.11 and 66.18 cases per 100,000 pregnancy months [36,37,38]. Our study focused on women in the immediate postpartum period. We found that the IgG+ seroprevalence rate was 28.6%, whereas the IgM+ seroprevalence rate was 31%. A follow-up study of IgG antibodies showed a median duration of 180 days, whereas with IgM, the median was 53 days [39]. It is estimated that most mothers who tested positive for IgG but negative for IgM antibodies were infected with Zika virus during the third trimester of pregnancy, considering the longevity of antibodies. Our group of patients acquired the infection between September and November 2017, which could indicate a low risk of birth defects as previously suggested [40]. For those who were IgG+/IgM+, the serology result could be associated with a ZIKV infection in the third trimester of pregnancy as well. However, in cases where individuals tested IgG+/IgM+, we cannot rule out the possibility of a reinfection. In those IgG+/IgM− cases, the infection could have occurred between May and September, with a higher probability of infection during the first or second trimester and, thus, a high risk of birth defects [41,42]. There is also evidence of antibody cross-reactivity between ZIKV and other flaviviruses, which increases the possibility that our serology is associated with the detection of anti-DENV [43]. Considering that the study site is an endemic DENV zone, coinfection between DENV and ZIKV could occur. However, DENV was not specifically assessed in this study. Additionally, during the immediate postpartum period, we found that 10.3% of the mothers had an active ZIKV infection detected by PCR. This suggests that the infection most likely occurred during the third trimester of pregnancy, unless long viremia occurred in some cases, which has been reported as well [44].

There is evidence to suggest that mothers with an active Zika virus (ZIKV) infection can transfer the virus to their newborns through breast milk (BM). In a previous study, blood–DBS tests on infants aged one year revealed that they had acquired a ZIKV infection through both horizontal and vertical transmission [45]. In our study on 126 mothers, 2 (1.5%) tested positive for ZIKV through a PCR analysis of their BM in the first 12 h after giving birth. Our study was limited to detecting viral ZIKV infection only during this immediate postpartum period; however, it is important to further investigate the presence of ZIKV in breast milk throughout lactation and determine its potential transmission of ZIKV particles to the newborn.

Previous studies have used DBS to test for antibodies against ZIKV and DENV viruses [18]. We chose this method because it is easy to transport from Oaxaca to Mexico City and was recommended by the ethics committee following international guidelines [46]. Out of a total of 126 newborns, 7 were found to be IgG-DBS-positive, meaning that 5.5% of the infant samples contained IgG antibodies (as shown in Table 3). These antibodies were transferred vertically from the mother, either after she was infected with ZIKV or perhaps after a DENV infection, which can cause a cross-reaction. Such cross-reactions have been observed and would be particularly beneficial because these infants would acquire antibodies to protect against vector-transmitted infections. Although these responses are low, they are sustainable and may offer postnatal protection [47,48,49].

It is highly unlikely that our infants could have been infected with ZIKV through mosquito bites as they had no contact with the environment. However, we observed that eight infants tested positive for ZIKV in their saliva at birth. This suggests that the transmission of ZIKV occurred vertically, through the placenta and other secretions. It is important to note that the fetus starts making sucking and swallowing movements around the 12th week of gestation, but can open their mouth as early as the 10th week, thus having early contact with amniotic fluid. This is one way the fetus could contract ZIKV-positive fluid. Although we did not test amniotic fluid for ZIKV virus, its presence has been detected in previous studies. Therefore, some of the infants that tested positive for ZIKV virus in their saliva may have acquired the virus through contact with amniotic fluid [18,50,51,52,53]. It has also been observed that ZIKV is more easily detected in saliva than in blood during the acute phase. Our results also confirm the previously observed perinatal transfer [51]. After analyzing all the infants, we found that three of them were PCR-positive in DBS, indicating active infection. Previous evidence suggests that PCR on the DBS of ZIKV can detect viral RNA for up to 180 days. In our case, the samples were transported immediately, which increased our confidence in the results [18].

To investigate these results further, we performed a microneutralization assay on those samples. In most cases, the highest titers were observed in IgG+ samples from the mother and newborn, as IgG antibodies come from the mother. The high neutralizing activity of the IgG antibodies from the mother provide evidence of protection for the infant, acquired through intrauterine transfer, to the ZIKV infection, acquired transplacentally [52,53]. Our work was limited to detecting neutralizing antibodies in the blood of the infant, which could protect the infant both intrauterine and extrauterine until their longevity ends. However, detecting neutralizing antibodies in milk is also necessary to determine the ability of breastfeeding to protect the infant during their extra uterine life for a long time. In most cases, low ED50 titers were observed, which were interpreted as no neutralization and correlated with the detection of IgM, which is known to be unable to cross the placental barrier. In some cases, IgG+/IgM+ showed a high ED50 level, while others showed a low ED50 level. This could mean that despite both being PCR-positive and therefore having active infections, the first cases were infections acquired previously during the pregnancy, whereas the second cases were more recent infections. In the first cases, the antibodies found were mature and had better avidity than in the second case (Figure 2) [52,53]. Moreover, we noticed that some infants who tested negative for IgG had a significant level of neutralization. Some studies suggest that low levels of IgM can pass through the placental barrier, and their concentration can increase in the case of inflammation [49]. Although we are not certain about any particular event, it is possible that the neutralizing activity is linked to the transfer of IgM during such an event.

We observed a high prevalence of microcephaly (percentile ≤ 10) in our study population (23 out of 126; 18.25%). We then conducted serological and molecular tests for IgG, IgM, PCR of the blood of the mother, and saliva PCR in the infant to determine if the microcephaly was associated with ZIKV infection. It is important to note that ZIKV infection damages the developing brain during the first and second trimesters of pregnancy [53]. Our findings suggest that saliva PCR had the strongest correlation (OR = 9.2, *p* = 0.0040), indicating that at least 5 out of 23 (21.7%) microcephaly cases were associated with ZIKV infection. This may be due to early contact between the fetus and amniotic fluid infected with ZIKV through the mouth, which could occur in the first two trimesters of pregnancy. IgG and IgM tests showed associations with microcephaly in 9 out of 23 (39%) and 14 out of 23 (60.7%) cases, respectively. This suggests that these antibodies may have been produced during an infection that occurred in the first two trimesters of pregnancy. We analyzed the neutralizing activity and microcephaly in six cases of infants who tested positive for IgG antibodies in their blood–DBS samples. Out of these six cases, two showed neutralizing antibodies, while the other four did not. The first case was IgG+/IgM−, suggesting that the infection was not recent and may have occurred in the first trimester of pregnancy as a primary infection. This could have had an impact on the fetus, resulting in the neutralizing activity from the antibodies of the infection, which were mature at birth. The second case may have had a recent reinfection, and the IgG antibodies had low [53].

## 5. Conclusions

Maternal IgG seroprevalence was 28.4% with 10.4% active infection, while infant IgG seroprevalence was 5.5% with 2.4% active infection. There were two cases of virolactia, and 6.3% of the infant saliva samples tested positive for ZIKV. Additionally, 18.3% of the infants were in a cephalic perimeter percentile lower than 10 and had an association between microcephaly and serology or a PCR between 8.6 and 60.9%. The infant blood samples had neutralizing antibodies, indicating intrauterine protection. Microcephaly was correlated with serology or PCR, but in our study population, non-ZIKV factors may be involved as well. Low ZIKV infection values in breast milk mean that breastfeeding is safe in most of the mothers and infants of the endemic area studied.

## Figures and Tables

**Figure 1 microorganisms-12-00423-f001:**
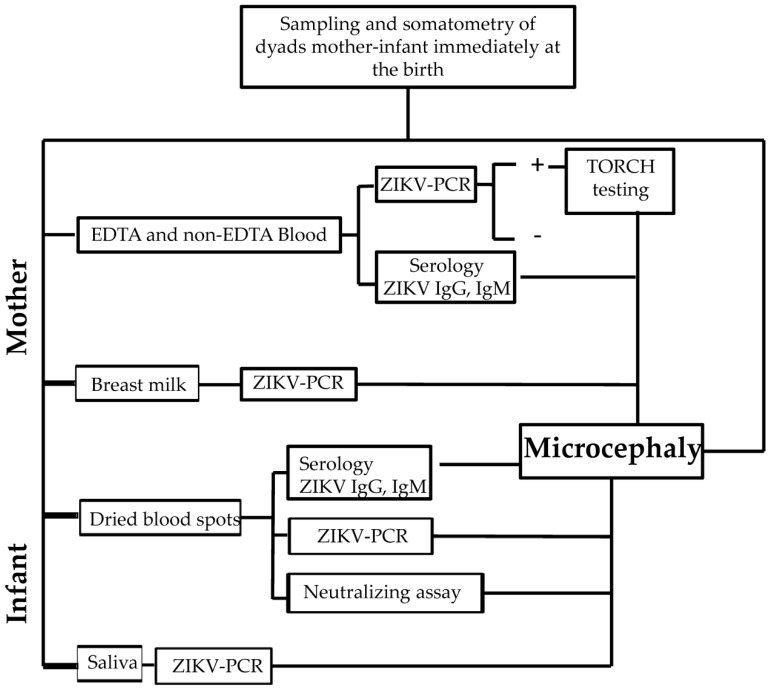
The flow chart of the experimental methodology carried out in the present study.

**Figure 2 microorganisms-12-00423-f002:**
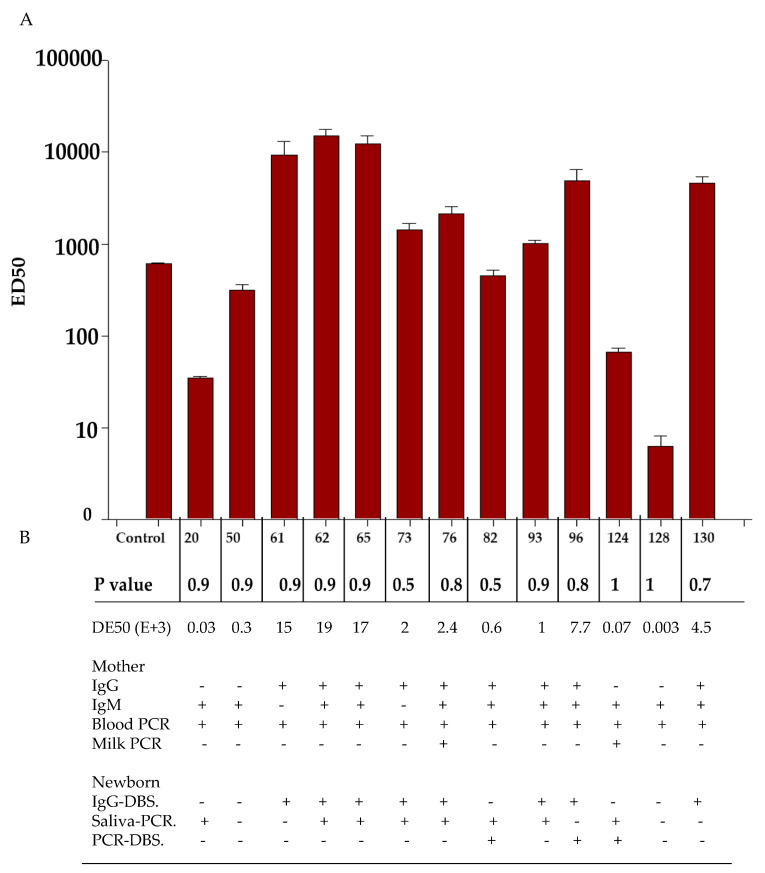
Analysis of the neutralization of blood samples of infants with serology and active infection at birth. (**A**) ED50 was obtained from 13 samples from infants that were positive (determined by PCR) which included 6 IgG-positive samples from the blood of the infants. (**B**) Samples from mothers and infants with active infection, comparing the DE50 with serology and PCR in the samples from the mothers and infants.

**Table 1 microorganisms-12-00423-t001:** Demographic characteristics of the population.

Characteristic	N	Median	Min	Max
Mother				
Age (years)	126	23	14	40
Infant				
Female	65 (52%)			
Male	61 (48%)			
Gestational age				
Preterm (<37 WG*)	4 (3%)			
To term (37–41 WG*)	120 (95%)			
Post-term (≥41 WG*)	2 (2%)			

WG*: weeks of gestation.

**Table 2 microorganisms-12-00423-t002:** Correlation between serology and PCR in maternal blood.

	IgG+/IgM+	IgG+/IgM−	IgG−/IgM+	IgG−/IgM−
PCR+	7/25 (28%)	1/6 (0.8%)	5/11(45.5%)	0
PCR−	18/25 (18.3%)	5/6 (4.8%)	6/11 (54.5%)	84/126 (66.6%)

**Table 3 microorganisms-12-00423-t003:** Correlation between the number of cases of microcephaly and antibodies or active infection in the mother and/or in the infant.

		MOTHER				INFANT					
		IgG		IgM		Blood—PCR		Saliva—PCR		IgG DBS	
		+	−	+	−	+	−	+	−	+	−
Microcephaly	YES	9	14	14	9	7	16	5	18	2	21
	NO	27	76	25	78	6	97	3	100	4	99

+: Positive, −: Negative.

**Table 4 microorganisms-12-00423-t004:** IgG-positive cases in the DBS samples from the infants and their relation to neutralization and head circumference.

Pair	HC *	Mother		RT PCR		Serology IgG	RT-PCR Infant		ED50
		IgG	IgM	Blood	BM	Infant	Saliva	DBS	
61	76	+	−	+	−	+	−	−	15 × 10^3^
62	85	+	+	+	+	+	+	−	19 × 10^3^
65	80	+	+	+	−	+	+	−	17 × 10^3^
73	6	+	−	+	−	+	+	−	2 × 10^3^
82	33	+	+	+	−	+	+	+	0.06 × 10^3^
93	6	+	+	+	−	+	+	−	1 × 10^3^
96	93	+	+	+	−	+	−	+	7.7 × 10^3^
130	75	+	+	+	−	+	−	−	4.5 × 10^3^

HC: * Head circumference, BM: breast milk, DBS: dried blood spots.

## Data Availability

Data are contained within the article.

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
