# Peer review of "Intrauterine Transmission of Zika and Vertical Transfer of Neutralizing Antibodies Detected Immediately at Birth in Oaxaca, Mexico: An Analysis in the Context of Microcephaly"

_microorganisms, 2024, doi:10.3390/microorganisms12030423_

Round 1
Reviewer 1 Report
Comments and Suggestions for Authors
The manuscript authored by Alfredo and colleagues presents findings on intrauterine transmission of Zika and transfer of neutralizing antibodies from the mother to the newborn.
Therefore, this manuscript by Alfredo et al. has the major findings:
“1) 28.4% of mothers had IgG seroprevalence, while only 5.5% of infants had IgG seropositivity;
2) The prevalence of active infection was 10%, and only two cases of virolactia were detected, indicating a low presence of ZIKV in breast milk;
3) ZIKV was detected in infant saliva, suggesting that it was acquired through the secretion of viral particles in amniotic fluid;
4) The study also found a good correlation between the detection of serology and PCR in different samples and microcephaly, with an association of 8.6 and 60.9%, respectively.
The authors' study is relevant in the context of ZIKV. However as highlighted below, overall, the introduction (background information), methodology and Discussion would benefit from a more coherent structure and concise explanation of the researches. Thus, I have several points that I see interesting to add to this manuscript:
1. Title (lines 1-3): Please considerer writing “Intrauterine Transmission of Zika Virus and Vertical Transfer of Neutralizing Antibodies Detected Immediately at Birth in Oaxaca, Mexico: An Analysis in the Context of Microcephaly”
2. Abstract (lines 35-48): Please considerer review the abstract that contains several grammatical errors.
3. Abstract (line 38): The acronym TORCH (ie, differential diagnosis for any sick neonate) is not clear to the reader. Rephrase this sentence since it is confusing.
4. Materials and Methods (line 92): Please considerer try to review this entire section, which contains several grammatical errors (for example: “from the infant, All samples”).
5. Materials and Methods: The methodology part was not clear. I suggest adding a flowchart with all the steps performed to make it clearer to the reader.
6. Materials and Methods: Again, I emphasize that this review needs to be carefully reviewed for grammatical and writing errors, for example (line 176): considerer writing “ZIKV” instead of “zika infection”.
7. Table 2 (line 186): The data in percentages is not clear: 77.7% of what? 22.2% of what? Make this clearer.
8. Discussion (line 260): considerer writing “DENV is transmitted” instead of “dengue is transmitted”.
9. Discussion (line 263): considerer writing “ZIKV infection” instead of “Zika virus infection”. For the entire text, systematize the acronyms. Once defined, use it systematically.
10. Discussion: Review several paragraphs that contain several grammatical errors, for example: ZIKV instead of ZIKV virus…
11. I emphasize that throughout the manuscript there are several conceptual errors, for example dengue and Zika refer to the disease caused by DENV and ZIKV, respectively. So it's wrong to say that "Coinfection of dengue and Zika may have occurred”.
12. Discussion: It is important for the authors to formulate a paragraph regarding the limitations of the study.
Author Response
Dear Reviewer
The authors of the paper titled 'Intrauterine Transmission of Zika and Vertical Transfer of Neutralizing Antibodies Detected Immediately at Birth in Oaxaca, Mexico: An Analysis in the Context of Microcephaly' have made the necessary corrections to our paper based on your observations. These corrections have allowed us to improve the paper, and we are grateful for your valuable feedback.
- Puctuation of the title was corrected , thanks for the observation. Now the title is: INTRAUTERINE TRANSMISSION OF ZIKA AND VERTICAL TRANSFER OF NEUTRALIZING ANTIBODIES DETECTED IMMEDIATLY AT THE BIRTH IN OAXACA, MEXICO: AN ANALYSIS IN THE CONTEXT OF MICROCEPHALY.
- Abstract (lines 35-48): Please considerer review the abstract that contains several grammatical errors.
To enhance the quality and understanding of the abstract, it has been rewritten.
- 3. Abstract (line 38): The acronym TORCH (ie, differential diagnosis for any sick neonate)
is not clear to the reader. Rephrase this sentence since it is confusing.
To avoid any lack of clarity, the abstract has been completely reformulated.
- Materials and Methods (line 92): Please considerer try to review this entire section, which
contains several grammatical errors (for example: “from the infant, All samples”).
The native speaker reviewed the grammatical errors and corrected most of the material and methods.
- Materials and Methods: The methodology part was not clear. I suggest adding a flowchart
with all the steps performed to make it clearer to the reader.
Thanks for your observation. We have added a flow chart to provide a clearer outline of our materials and methods section.
- Materials and Methods: Again, I emphasize that this review needs to be carefully reviewed for grammatical and writing errors, for example (line 176): considerer writing “ZIKV” instead of “zika infection
The methodology was improved and several sections were rewritten. The grammar was corrected with the help of a native speaker.
- Table 2 (line 186): The data in percentages is not clear: 77.7% of what? 22.2% of what? Make this clearer.
Table 2 was removed, and all percentage values were moved to the text. Now, with the new form of presentation, the percentages are clearer.
- Discussion (line 260): considerer writing “DENV is transmitted” instead of “dengue is transmitted”.
The discussion was rewritten, taking into account this observation.
- Discussion (line 263): considerer writing “ZIKV infection” instead of “Zika virus infection”. For the entire text, systematize the acronyms. Once defined, use it systematically.
The entire discussion was revised, with particular attention given to acronyms and overall clarity of the text.
- Discussion: Review several paragraphs that contain several grammatical errors, for example: ZIKV instead of ZIKV virus…
confirm that a native speaker has reviewed all the text.
- I emphasize that throughout the manuscript there are several conceptual errors, for example dengue and Zika refer to the disease caused by DENV and ZIKV, respectively. So it's wrong to say that "Coinfection of dengue and Zika may have occurred”.
After reviewing the manuscript, we corrected and rewrote it to eliminate conceptual and grammatical errors.
- Discussion: It is important for the authors to formulate a paragraph regarding the limitations of the study.
Thanks for this observation. Several sections of the manuscript were reworded, and the limitations of the study were clarified where necessary in this new version.
Reviewer 2 Report
Comments and Suggestions for Authors
This work is interesting, but it can be improved, particularly in English revision...I think that some sentences have to be rewritten...
After this, tables and figures must be improved.
Table 1...rewrite and improve to better visualization
Table 2...not necessary
Tables 4 and 5...difficoult to understand...too numbers and several abbreviations
Figure 1...improve...numbers are too little
Comments on the Quality of English Languagemistakes throughout the entire manuscript and in some case there are sentences that have not been correctly written
example:
Briefly: Serum samples from the mother diluted 1:10 in dilution buffer of sample and controls were added in each well followed by HRP-conjugated, incubated at 37°C and then washed 4 times. Afterward, chromogen solution was added and incubated 15 min/37°CCA, USA) finally stop solution was added and reed to 450 nm in an ELISA reader. Whereas maternal IgM was done like above but with the kit by the MyBioSource Zika IgM kit (cat. MBS109003, San Diego Cal). Whereas ELISA IgG was done in DBS of the newborns by the use of the Euroimmune kit (cat EI2668-9601G, Lübeck, Germany) and according the previously published [16].
Author Response
The authors of the paper entitled "Intrauterine Transmission of Zika and Vertical Transfer of Neutralizing Antibodies Detected Immediately at Birth in Oaxaca, Mexico: An Analysis in the Context of Microcephaly" are grateful for the observations that have allowed us to improve the quality of the paper. Corrections have been made as follows:
After this, tables and figures must be improved. Most of the was rewritten additionally the English was reviewed by a native speaker
Table 1...rewrite and improve to better visualization. We are grateful with this observation.
Response: To aid comprehension, Table 1 has been simplified.
Table 2...not necessary.
Response: The content of Table 2 was integrated into the text and the table itself was removed
Tables 4 and 5...difficoult to understand...too numbers and several abbreviations
Response: Tables 4 and 5 were simplified and improved, and the meanings of acronyms were included as figure captions.
Figure 1...improve...numbers are too little
Response: We have made a new version of the presentation, where Figure 1 has been replaced with Figure 2.Comments on the Quality of English Language. The English language was review it by a native speaker
Round 2
Reviewer 1 Report
Comments and Suggestions for Authors
The authors have followed the suggestions and improved the manuscript.
Reviewer 2 Report
Comments and Suggestions for Authors
The revision is good...thank you